# Biomarkers and Data Visualization of Insulin Resistance and Metabolic Syndrome: An Applicable Approach

**DOI:** 10.3390/life14091197

**Published:** 2024-09-21

**Authors:** Christos Sotiropoulos, Nikolaos Giormezis, Vayianos Pertsas, Theodoros Tsirkas

**Affiliations:** 1Department of Microbiology, School of Medicine, University of Patras, 26504 Patras, Greece; christos.sotiropoulos@gmail.com; 2Informatics Department, University of Economics and Business, 10434 Athens, Greece; vpertsas@aueb.gr; 3Independent Researcher, 10434 Athens, Greece; thodoristsirkas@gmail.com

**Keywords:** insulin resistance, metabolic syndrome, BMI categorization, NHR Index, data visualization

## Abstract

Type 2 diabetes, prediabetes, and insulin resistance (IR) are widespread yet often undetected in their early stages, contributing to a silent epidemic. Metabolic Syndrome (MetS) is also highly prevalent, increasing the chronic disease burden. Annual check-ups are inadequate for early detection due to conventional result formats that lack specific markers and comprehensive visualization. The aim of this study was to evaluate low-budget biochemical and hematological parameters, with data visualization, for identifying IR and MetS in a community-based laboratory. In a cross-sectional study with 1870 participants in Patras, Greece, blood samples were analyzed for key cardiovascular and inflammatory markers. IR diagnostic markers (TyG-Index, TyG-BMI, Triglycerides/HDL ratio, NLR) were compared with HOMA-IR. Innovative data visualization techniques were used to present metabolic profiles. Notable differences in parameters of cardiovascular risk and inflammation were observed between normal-weight and obese people, highlighting BMI as a significant risk factor. Also, the inflammation marker NHR (Neutrophils to HDL-Cholesterol Ratio) Index was successful at distinguishing the obese individuals and those with MetS from normal individuals. Additionally, a new diagnostic index of IR, combining BMI (Body Mass Index) and NHR Index, demonstrated better performance than other well-known indices. Lastly, data visualization significantly helped individuals understand their metabolic health patterns more clearly. BMI and NHR Index could play an essential role in assessing metabolic health patterns. Integrating specific markers and data visualization in routine check-ups enhances the early detection of IR and MetS, aiding in better patient awareness and adherence.

## 1. Introduction

Insulin resistance (IR) is a common metabolic disorder in the developed world. Its pathogenic mechanism is the inability of muscle, fat, and liver cells to respond adequately to the action of insulin, making glucose uptake difficult. IR could cause type 2 diabetes, hypertension, dyslipidemia, endothelial dysfunction, and non-alcoholic fatty liver disease (NAFLD) [1]. It is estimated that at least 15.5% of the European [2] and 40% of the American population have IR and that it takes approximately 10–15 years to develop into type 2 diabetes [3,4]. Before reaching that point, there is a stage between IR and diabetes called prediabetes. This is a condition where pancreatic cells cannot compensate for the body’s insulin needs caused by IR. The prevalence of prediabetes is high (15.8–20.2%). About 25% of individuals will develop diabetes in 3–5 years, while 70% will develop it at some point in their life [5].

At the same time, the aforementioned spectrum of glucose homeostasis disturbances is part of another equally serious medical condition: Metabolic Syndrome (MetS). Two of the most serious consequences of MetS are the increased chances of a serious cardiovascular event (ASVD) [6,7] and the presence of chronic inflammation in individuals [8]. Its prevalence is estimated that >50–70% of the general population (estimated >90% in the Greek population) know little to nothing about the existence of MetS as a pathological entity [9]. However, the diagnosis of the aforementioned diseases is not as easy as it seems. The clinicians suspect prediabetes from marginally high fasting glucose levels in consecutive measurements. The diagnosis of IR is even more difficult since it cannot be detected by basic laboratory tests or personal and family history, but requires specialized tests that must be specifically requested, which are expensive or/and robust. Additionally, although the recognition of MetS is based on easily performable laboratory and anthropometric measurements, its diagnostic criteria are rarely used in daily clinical practice due to the lack of consensus about its usefulness.

In Greece, private hematological/biochemical laboratories conduct screenings of basic metabolic parameters (glycemic-lipemic-hepatic profile, complete blood count) in the general population through annual check-ups. The results are given to the individuals in a simple format where the presentation of each biochemical parameter is limited to the given value and the reference range. In fact, this format has not changed at all for at least the last 40 years, so the question now arises whether the information provided is incomplete or not. Also, there are no satisfactory laboratory algorithms for recognizing IR and MetS. These can lead to poorer patient-centered medical care and decreased patient–clinician relationships [10].

Therefore, these laboratories are the most suitable environment to investigate the following: (a) the prevalence of IR and MetS in a community-based population, two diseases that are not detected by traditional diagnostic methods; (b) the recognition of basic metabolic profiles; (c) the performance of existing diagnostic markers of IR in daily laboratory practice; and (d) the help that can be offered in informing and addressing IR and MetS.

## 2. Materials and Methods

### 2.1. Study Design

Initially, the study included 2274 individuals, which, after the application of exclusion and inclusion criteria, reduced to 1870 individuals. A group of patients with MetS was created, whereas individuals without MetS were further divided into 3 subgroups using BMI (Body Mass Index) as the primary factor: normal-weight individuals (BMI < 25), overweight individuals (BMI 25–30), and obese individuals (BMI > 30). Indices of cardiovascular danger, inflammation, and glucose homeostasis were defined in the above groups in order to investigate whether this categorization creates distinct metabolic profiles, which could potentially contribute to the personalized interpretation of the results. For this reason, a data visualization format was created, and its acceptance was recorded. Furthermore, the BMI-based groups were used to check the performance of indices for the diagnosis of IR in the Greek population. The cut-off of HOMA-IR, the most commonly used index for IR diagnosis, was determined for each group, and the validity of low-budget IR indices such as the TYG Index, TyG-BMI Index, and Trig/HDL ratio was also tested in order to evaluate their use in everyday practice (Figure 1).

### 2.2. Refining the Study Population Using Exclusion and Inclusion Criteria

The criteria aimed to (a) exclude both severe or active physiological and pathological conditions that significantly contribute to sugar, lipid, and immune system disorders; (b) exclude conditions that tend to create outliers, such as individuals with extreme height and weight values; and (c) focus on the segment of the population with the most common leisure-time physical activity (LTPA) profile found in Greece (Table 1). It should be noted that hypothyroidism was not in the exclusion criteria due to its high incidence in the Greek population (9% in the general population and 14.4% in women [11]) and also due to its association with insulin resistance [12].

### 2.3. Selection of BMI as the Primary Diagnostic Factor for Metabolic Syndrome (MetS) and Insulin Resistance (IR), as well as the Creation of Groups with Distinct Metabolic Profiles

At the initial stages of our study, some individuals undergoing check-ups showed discomfort or refusal to have their waist circumference (WC) measured, while they were more receptive to determining their BMI. Therefore, the study decided to rely solely on BMI for the analysis. BMI is the most common tool for categorizing obesity and has been utilized in numerous clinical and research studies while being considered a risk factor due to its correlation with various diseases [13,14,15,16]. Particularly regarding the prediction, diagnosis, and management of MetS and IR, the association with BMI is very strong [17,18,19,20,21]. Additionally, BMI thresholds and categories are non-ethnicity-specific and have been well-defined and accepted for several decades. In contrast, WC does not have fixed categories, only ethnicity-specific cut-offs [22]. Another key advantage of using BMI for determining weight-related health risks is that the measures needed for its calculation—height and weight—can be taken with high accuracy when performed using standardized procedures [23].

The main disadvantage of BMI is that, unlike WC, it does not accurately estimate abdominal obesity, which plays an equally or even more significant role than BMI in various diseases, such as MetS and CVDs [24,25,26,27]. In our study, we attempted to mitigate this drawback by excluding individuals with extreme height and weight values, as well as those with intense physical activity, thus reducing the likelihood of false-positive categorization [23]. It should be noted that BMI was calculated as follows: each individual was asked to stand upright wearing pants, a short-sleeved or long-sleeved shirt, and without shoes on a scale with an integrated height meter. Measurements were taken twice, and the mean value was chosen. BMI was calculated using the Quetelet index formula (BMI = weight/height^2^).

### 2.4. Geographical, Ethnological, and Social Characteristics of the Community of Interest

The study included individuals who visited our laboratory (Private laboratory of biochemistry and microbiology, primary health care) for check-ups during the period from September 2022 to February 2024. These individuals were all of Greek nationality and resided in the center of Patras, a coastal city that is the largest in southern Greece. Approximately 25% were retirees, 65% were employed, and 10% were students. Around 76% of the employed individuals were public or private sector employees, 5% were involved in manual labor, and approximately 19% were employers. None were engaged in agriculture. Τhe study was conducted in accordance with the Declaration of Helsinki (as revised in 2013), as well as with the legislation of the Greek National Commission for Bioethics and Technoethics.

### 2.5. Final Refining the Study Population Using 

Initially, the study included 2174 individuals, which after the application of exclusion and inclusion criteria, reduced to 1870 individuals. The criteria aimed to (a) exclude both physiological and pathological conditions (with low prevalence) that significantly contribute to sugar, lipid, and immune system disorders, (b) exclude conditions that tend to create outliers, and (c) focus on the segment of the population with the most common leisure-time physical activity (LTPA) profile found in Greece [28]. Information regarding the criteria was obtained from a questionnaire administered orally at the time of blood collection, either to the individuals themselves or to accompanying relatives.

### 2.6. Identification of Individuals with Metabolic Syndrome

The diagnostic criteria for MetS primarily followed the WHO guidelines from 1998 [29]. Additionally, to reduce the likelihood of false negatives, the NCEP ATP III definition (revision 2005) was used, allowing for the diagnosis of MetS in individuals without using waist circumference criteria but by meeting three of the following four criteria: hyperglycemia, hypertriglyceridemia, low HDL, and hypertension. It should be noted that hypertension was identified based on self-reports.

### 2.7. Identification of Individuals with Insulin Resistance 

HOMA-IR was used as the reference method for diagnosing IR, calculated as insulin (mU/L) × glucose (mg/dL)/405. The selection of the HOMA-IR cut-off was based on the study of percentiles from the distribution of HOMA-IR values in individuals considered to have the healthiest metabolic profile, characterized by the following: BMI < 25, aged 18–64 years without prediabetes or type 2 diabetes, without hypothyroidism, and without MetS—factors that unpredictably alter HOMA-IR levels.

### 2.8. Key Biochemical and Hematological Parameters Studied—Comparison of Biochemical and Hematological Parameters between the Study Groups

Four categories of parameters were evaluated in the MetS (+) group and the BMI subgroups: (a) Parameters primarily acting as cardiovascular risk factors: Total Cholesterol (TC), Triglycerides (TG), HDL cholesterol, LDL cholesterol, non-HDL cholesterol, TC/HDL ratio, non-HDL/HDL ratio, and ALT/AST ratio. (b) Parameters primarily serving as inflammation indicators: White Blood Cell count (WBCs), Neutrophil count (NEUs), Lymphocyte count (LEUs), and NEUs/LEUs ratio (NLR Index). (c) Parameters serving as indicators for both the above pathologies: NHR Index (Neutrophils to HDL-Cholesterol Ratio) and hs-CRP. (d) Parameters primarily serving as indicators of glycemic status: Glucose, TG/HDL ratio, and insulin (Table 2).

Although insulin and hs-CRP are not part of routine check-ups, they were deemed necessary for our study. The former is for HOMA-IR calculation and confirmation of glucose disorders. The latter is due to its high correlation with IR, chronic inflammation, and CVDs [30,31]. Since the majority of parameters had non-normal distributions, their values were presented as median + interquartile range (IQR). Subsequently, using the variation in the aforementioned parameters across different groups, an attempt was made to identify key desirable and undesirable metabolic health patterns through machine-learning classification.

Complete blood counts were conducted using the Sysmex XN-300 hematology analyzer. However, biochemical and hormonal parameters were analyzed within 24 h of serum separation using the HORIBA Pentra C400 and SNIBE Maglumi analyzers. Lastly, the performance of low-cost diagnostic markers of IR was evaluated using HOMA-IR as a reference method. The following indices were studied: TyG Index [32], TyG-BMI Index [33], Triglycerides/HDL ratio, and ALT/AST ratio.

**Table 2 life-14-01197-t002:** Reference values, units, and laboratory methods of each parameter. * LDL = Total Cholesterol—HDL—(Triglycerides/5). ** non-HDL = Total Cholesterol—HDL. *** TyG Index = Ln (fasting triglycerides [mg/dL] × fasting glucose [mg/dL])/2. **** TyG-BMI Index = TyG Index × BMI.

	Reference Value	Units	Laboratory Method
White Blood Cells (WBCs)	4000–10,000	count/μL	Flow cytometry
Neutrophils (NEUs)	1500–6500	count/μL	Flow cytometry
Lymphocytes(LEUs)	1500–3500	count/μL	Flow cytometry
Glucose	70–110	mg/dL	nephelometry
Total Cholesterol	140–200	mg/dL	nephelometry
Triglycerides	<150		nephelometry
HDL Cholesterol	>40 (men) > 50 (women)	mg/dL	nephelometry
LDL Cholesterol	<116	mg/dL	* calculated
Non-HDL Cholesterol	<145	mg/dL	** calculated
ALT	<38	U/L	Nephelometry
AST	<40	U/L	Nephelometry
Insulin	3–25	μU/mL	CLIA
Hs-CRP	<0.03	mg/dL	CLIA
TyG Index	4.49–4.83 [34]	none	*** calculated
TyG-BMI Index	191–199 [35]	none	**** calculated
Triglycerides/HDL	<2.1	none	Calculated
Neutrophils/Lymphocytes (NLR) Index	Comparison between groups	none	Calculated
ALT/AST ratio	066–0.70	none	Calculated
Neutrophils/HDL (NHR) Index	Comparison between groups	none	Calculated

### 2.9. Creation and Evaluation of a New Diagnostic Index for Insulin Resistance (IR)

Initially, a Spearman rank correlation was performed on 26 numerical and non-numerical parameters with HOMA-IR in individuals without MetS. Subsequently, the possible use of 12 parameters with the greatest relevance to HOMA-IR in constructing a new IR diagnosis index via logistic regression analysis was studied. After trials, the following parameters were selected: BMI, Glucose, Triglycerides, and NHR Index. To facilitate reading and understanding of the work, the new index was named “Patraslab Index”. Its equation was created with 75% of the individuals without MetS as the training group. The index’s performance was tested in the training group along with the other low-budget IR indices. Furthermore, the performance in association with IR prevalence in the general population was studied. To this end, we created variants of evaluating groups with different prevalence of IR from the remaining 25% of individuals without MetS. Lastly, the sensitivity and specificity of the PatrasLab Index were checked in these groups.

### 2.10. Blood Sampling Conditions

Blood samples were collected in the morning hours (08:00–10:00 a.m.) in a specially designated area within our laboratory. The blood collection procedure followed the guidelines outlined by the World Health Organization (WHO) [36]. Participants were instructed to fast for 10–12 h and abstain from alcohol consumption for 24 h prior to the blood draw. All subjects gave their informed consent for inclusion before they participated in the study. The study was conducted in accordance with the Declaration of Helsinki. Informed consent was obtained from all subjects involved in the study.

### 2.11. Data Visualization

The parameters of glycemic and lipid profiles, as well as the two insulin resistance (IR) indices (TyG Index and TG/HDL ratio), were presented in a distinct format concurrently with the traditional presentation method. Specifically, their values were depicted using bullet graphs with three gradations: optimal (green), borderline (orange), and pathological (red). Additionally, the criteria for Metabolic Syndrome (MetS) were illustrated via a vertical stacked bar chart. Each parameter was accompanied by a concise explanatory/descriptive statement. A questionnaire was sent via email to all participants along with the lab results in the classic and the new visualized form. The majority of participants were 18–50 years old, as this age group preferred to receive the results via mail.

### 2.12. The Use of the ‘Rule-In’ Diagnosis Approach

The aim of the annual check-up is to screen the general population, and it is optional. Therefore, it has a cost-effective orientation. Hence, in our study, we prioritized diagnostic indices with as few false-positive results as possible to avoid increasing the cost of further investigation due to them [37]. This was ensured by setting a priority for them to have very high specificity and/or Positive Predictive Value (PPV) to confirm (rule in) a diagnosis.

### 2.13. Statistical Analysis

All statistical analyses were performed using the open-source software JASP (version 0.18.3 created by the University of Amsterdam). Since the majority of parameters had non-normal distributions, their values were presented as median + interquartile range (IQR). For the evaluation of the low-cost IR indices, ROC analysis was used. For the creation and evaluation of the PatrasLab Index, Spearman rank correlation analysis, machine learning analysis (random forest), and ROC analysis were used.

## 3. Results

### 3.1. Demographics and Medical Characteristics of the Study Population and Its Categorization

The basic characteristics of the selected population are as follows: 42.6% of individuals were male and 57.4% female. Young adults and young middle-aged individuals (18–44 years old) comprised 40.2% of the studied population, middle-aged individuals (45–64 years old) 31.2%, and elderly individuals (65–84 years old) 28.5%. Furthermore, 32.1% were chronic smokers. Additionally, the existence of chronic diseases with high prevalence contributing to the development of MetS and/or IR was included in the study. Thus, 25.8% reported receiving treatment for hypertension, 36.3% for acquired dyslipidemia, 27.1% for type 2 diabetes mellitus, and 21.1% for Hashimoto’s thyroiditis or non-immune hypothyroidism.

Regarding categorization, initially, two groups were created: individuals with MetS and individuals without MetS. In this regard, 16.2% of individuals were found to have MetS. Subsequently, individuals without MetS were further divided into three subgroups based on BMI: normal-weight individuals (BMI < 25 kg/m^2^), overweight individuals (BMI 25–30 kg/m^2^), and obese individuals (BMI > 30 kg/m^2^). These groups differed slightly to significantly in the aforementioned characteristics, with the exception of smoking status, which was nearly the same across all three groups.

On the other hand, differences between patient groups were observed. Most people with MetS were men in contrast to other groups, whereas the prevalence of hypothyroidism was high in individuals with MetS and BMI > 30. On the contrary, individuals with ΒΜΙ < 25 showed the lowest percentages of hypertension, dyslipidemia, type 2 diabetes, and hypothyroidism (Table 3).

### 3.2. Comparison of Biochemical Parameters between the MetS Group and BMI Groups, and the Creation of a Machine-Learning Model for Identifying Key Metabolic Health Patterns

The comparison of the basic metabolic profile of the studied groups is shown in Table 4, Table 5 and Table 6. The group with MetS had the most pathological profile of all, as almost all indicators were statistically significantly higher compared to the other groups, although the NLR Index and hs-CRP did not differ significantly between individuals with Metabolic Syndrome (MetS (+) and those with obesity (BMI > 30, MetS (−). On the other hand, individuals with normal weight (BMI < 25) and MetS (−) had the most normal profile of all, as almost all indices were statistically significantly lower compared to the rest. Notably, the NHR Index differed greatly between individuals with MetS (+) and those with normal weight (95.8 vs. 52.7).

Finally, an attempt was made to develop a model for identifying Metabolic Syndrome using machine-learning classification (Random Forest algorithm). The model was based on parameters such as age, gender, Glucose, Triglycerides, NHR Index, and ALT/AST ratio, which differed statistically significantly among all study groups (MetS (+) group and BMI subgroups). The resulting model could reliably identify individuals with MetS (AUC 0.85, F1 Score 0.84)

### 3.3. Determining the HOMA-IR Cut-Off

Individuals with normal weight without prediabetes or type 2 DM, hypothyroidism, and MetS were considered the healthiest in the study and, thus, with the lowest likelihood of having IR at the time of examination. Therefore, the 90th percentile of the HOMA-IR distribution was selected as the cut-off due to both the high specificity ensured by this percentile [38] and the expected prevalence of IR in individuals with BMI < 25. The value of the resulting cut-off was >2.78. The conclusion was that 79.8% of individuals with MetS and 28.9% without MetS had IR. Furthermore, 55.7% of individuals with BMI > 30, 34.2% with BMI 25–30, and 12.1% with BMI < 25 had IR.

### 3.4. Performance of Low-Cost Diagnostic Markers of IR

The sensitivity and specificity of the TyG Index [39], TTyG-BMI Index [40], Triglycerides/HDL ratio [41], ALT/AST ratio, Neutrophils/Lymphocytes ratio (NLR Index), Neutrophils/HDL ratio (NHR Index) were evaluated in the group of individuals without MetS (Table 7). The TyG-BMI Index had the best performance (AUC 0.77). The TyG Index and Triglycerides/HDL ratio had relatively good reliability (AUC 0.72 and 0.69, respectively). The ALT/AST ratio and NHR Index had moderate reliability, while the NLR Index had the worst performance.

Additionally, an attempt was made to create a new IR Index. After trials, the following parameters were selected: BMI, Glucose, Triglycerides, and NHR Index. To facilitate reading and understanding of the work, the new index was named “Patraslab Index”. Its performance was evaluated using ROC curve analysis. To determine sensitivity and specificity, priority was given to minimizing the false-positive rate. Therefore, emphasis was placed on achieving a high Positive Predictive Value (PPV), considering the prevalence of insulin resistance (IR).

Initially, the PatrasLab Index, along with the TyG-BMI Index, were tested in the training group (75% of individuals without MetS), and it emerged that the PatrasLab Index had the best performance (AUC 0.84). Subsequently, the indexes were tested in three variants of the control group with varying prevalence of insulin resistance (IR): low (10%), moderate (20%), and high (30%) IR prevalence. It was observed that the PatrasLab Index had better performance than the TyG-BMI Index in all versions, especially in populations with low (AUC 0.91, PPV 48.1, NPV 96.6) and high prevalence (AUC 0.81, PPV 68.2, NPV 85.3) of IR (Table 8).

### 3.5. Assessment of Individuals’ Response to Data Visualization

A questionnaire was sent via email to 486 individuals, of whom 263 responded. In total, 97.3% of respondents considered that presenting the parameters in the form of bullet graphs helped them understand their test results ‘well’ and ‘very well’, while over 90% found the explanatory statement accompanying each parameter very useful. Additionally, the majority of respondents knew ‘slightly’ and ‘not at all’ about the concept or existence of non-HDL cholesterol, insulin resistance, and Metabolic Syndrome (66.7%, 55.6%, and 67.8%, respectively), and they believed that this presentation format helped them better understand the role and value of these parameters (’very well’ and ’well’ responses 90.5%, 90.4%, and 88.9%, respectively). Finally, 95.3% of respondents considered that data visualization helped them have a more comprehensive understanding of their metabolic health profile (Figure 2).

## 4. Discussion

In Greece, it is estimated that there are approximately 2500 small to medium-sized private hematology/biochemistry laboratories primarily tasked with conducting annual check-ups. However, this check-up can be deemed inadequate since it does not include any information regarding IR and MetS, conditions that, based on their prevalence, affect a large portion of the general population. Furthermore, the current method of presenting results lacks important information, such as borderline values, which, in some cases, are highly valuable for the correct interpretation of those results (for example, in prediabetes). Our study addressed these concerns and explored ways to overcome them in everyday medical practice. The initial goal of our study was to identify distinct metabolic patterns during a routine health check-up. For this reason, a metabolic profile was created from inexpensive parameters, which included markers for cardiovascular risk and inflammation as well as for glycemic status. This profile was then studied across three BMI categories as well as in individuals with MetS. It was found that individuals with MetS exhibited the most adverse biochemical metabolic profile (in terms of cardiovascular risk factor, chronic inflammation, and glucose homeostasis) compared to all other groups. This was expected since MetS is associated with cardiovascular complications επιπλοκές [42] and chronic systemic infection [43], increasing the all-cause mortality [44]. Therefore, early diagnosis through low-cost parameters in routine laboratory practice can be essential in managing MetS. To this end, our study developed a machine-learning model that demonstrated exceptional performance in diagnosing Metabolic Syndrome. Also, as far as we know, this is the first time that an inflammation marker (NHR Index) has been integrated into a model for diagnosing Metabolic Syndrome. This suggests that machine-learning algorithms have the potential to significantly enhance the accuracy of identifying Metabolic Syndrome and may complement traditional diagnostic criteria.

On the other hand, BMI-based categorization showed that individuals with normal weight (BMI < 25) had the most favorable health profile of all, especially compared to individuals with MetS. Regarding overweight and obese individuals, the majority of parameters were the same in both BMI categories. Hence, it can be asserted that a BMI < 25 alone may serve as a protective factor against metabolic syndrome and cardiovascular diseases, consistent with the findings of previous studies [45]. It is noteworthy that the identification of metabolic patterns can be effectively accomplished by the machine-learning model developed in our study. Particularly for diagnosing Metabolic Syndrome, this model demonstrated exceptional performance. Also, as far as we know, this is the first time that an inflammation marker (NHR Index) has been integrated into a model for diagnosing Metabolic Syndrome. This suggests that machine-learning algorithms have the potential to significantly enhance the accuracy of identifying Metabolic Syndrome and may complement traditional diagnostic criteria.

Our study further aimed to explore the feasibility of using IR diagnostic indices in routine medical practice. While IR can be easily and reliably calculated using the HOMA-IR, this method requires insulin testing, which prescribing physicians seldom ask, mainly due to its cost. For these reasons, less expensive indices were evaluated. The indices with the best performance were the TyG and TyG-BMI Index, but their cut-offs differed from the proposed ones in the international literature (4.57 vs. 4.49 and 122 vs. 191 in participants without MetS, respectively). This indicates that such indices should always be validated for their performance before being used in a community-based laboratory.

Additionally, a new IR diagnostic index based on BMI, Glucose, Triglycerides, and NHR Index was created in our study. A strong correlation between Triglyceride, HDL, and Neutrophil count with insulin resistance has already been well reported [46,47]. On the other hand, as far as we know, this is the first time that the NHR (even as part of a formula) has been used for diagnosing IR. Consequently, the new index had much better characteristics than commonly recommended indices. Therefore, IR indices derived from the laboratory’s database, where they are intended to be used, are the most reliable and accurate.

Lastly, the usefulness of visualizing biochemical parameters was investigated. As far as we know, this is the first time that the visualization of medical data has been integrated into the results of an annual laboratory check-up. Its usefulness was evidenced by the positive feedback from participants. Specifically, the explanatory description of each test and its presentation in bullet graph format (with three value scales) were considered extremely valuable for understanding their overall metabolic health profile, which could enhance their health management [48]. At the same time, it was found that the general population has insufficient knowledge even about the most common biochemical tests. This was especially true for non-HDL cholesterol, which forms the basis for cardiovascular risk stratification (HellenicSCORE II+) [49], where only a small percentage of participants were aware of its significance or even its existence. The same was true for IR and MetS. Therefore, the visualization of biochemical tests proves essential to consider that a basic check-up fulfills its purpose. Also, it is deemed necessary as it ensures individuals are well-informed about their personalized metabolic profile, which leads to the reduction of information asymmetry between physicians and patients. In turn, this increases self-care and patient adherence to treatment [50].

## 5. Conclusions

Our study showed that low-cost biochemical parameters in a check-up can be utilized to accurately create distinct metabolic profiles, especially in people with MetS and individuals with ΒΜΙ < 25. Additionally, our study confirmed that the NHR Index is a significant risk factor, as has been shown in other studies [51]. At the same time, it appears to be a useful tool for diagnosing insulin resistance (IR) and Metabolic Syndrome (MetS). Therefore, it is worth exploring and utilizing further in studies of metabolic diseases. On the other hand, this study demonstrated that the results individuals receive during an annual check-up could be considered outdated, both in terms of the parameters included and their presentation. The improvement of both the content and presentation of annual check-up results, as well as optimization, is evident. Moreover, the ongoing evolution of laboratory practices necessitates the development of new methodologies, particularly those leveraging advanced technologies like machine-learning algorithms, to ensure accuracy, efficiency, and relevance in healthcare diagnostics [52]. Also, it is deemed necessary as it ensures individuals are well-informed about their personalized metabolic profile, which leads to the reduction of information asymmetry between physicians and patients [53]. In turn, this increases self-care and patient adherence to treatment.

### Limitations of the Study

The omission of WC increases the likelihood of misdiagnosing MetS despite counteraction from the use of exclusion and inclusion criteria. Nevertheless, the utility of easily measurable and manageable parameters such as BMI (in contrast to WC) prevails, as one of the purposes of this study is to facilitate an applicable method for identifying individuals with Metabolic Syndrome (MetS) in the everyday practice of a community-based laboratory. Moreover, the self-reporting of hypertension may also increase the likelihood of misdiagnosing MetS. However, it exhibits high specificity, resulting in a high Positive Predictive Value (PPV) for identifying individuals with Metabolic Syndrome from a community-based laboratory, thus aligning with the ‘rule-in’ diagnosis approach. On the other hand, both the new IR index proposed by our study and the data visualization tools need to be tested in other community-based laboratories to confirm their usefulness, something we intend to do in our next study. Finally, our cross-sectional study cannot determine whether providing individuals with enriched check-up information motivates them to follow a healthier lifestyle in the long term. For this reason, we plan to send participants a follow-up questionnaire annually for the next 3 years, and its results will be published in our subsequent study.

## Figures and Tables

**Figure 1 life-14-01197-f001:**
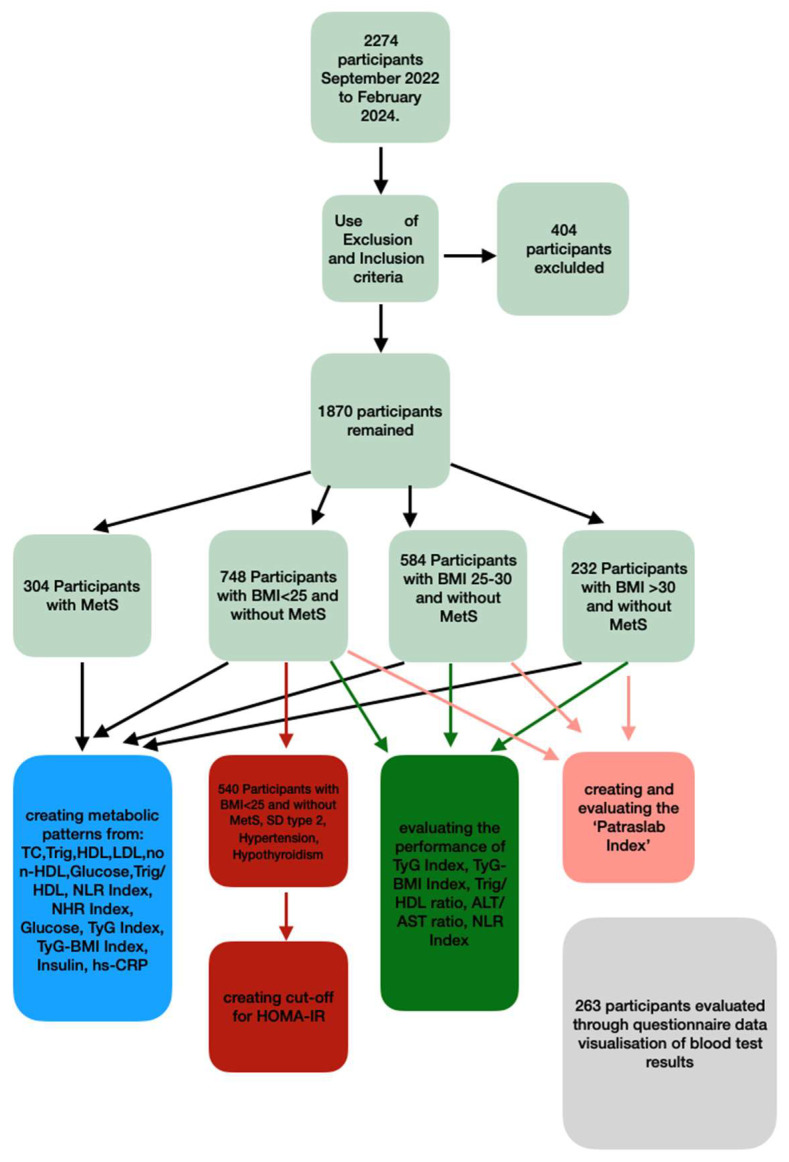
Flowchart of study design.

**Figure 2 life-14-01197-f002:**
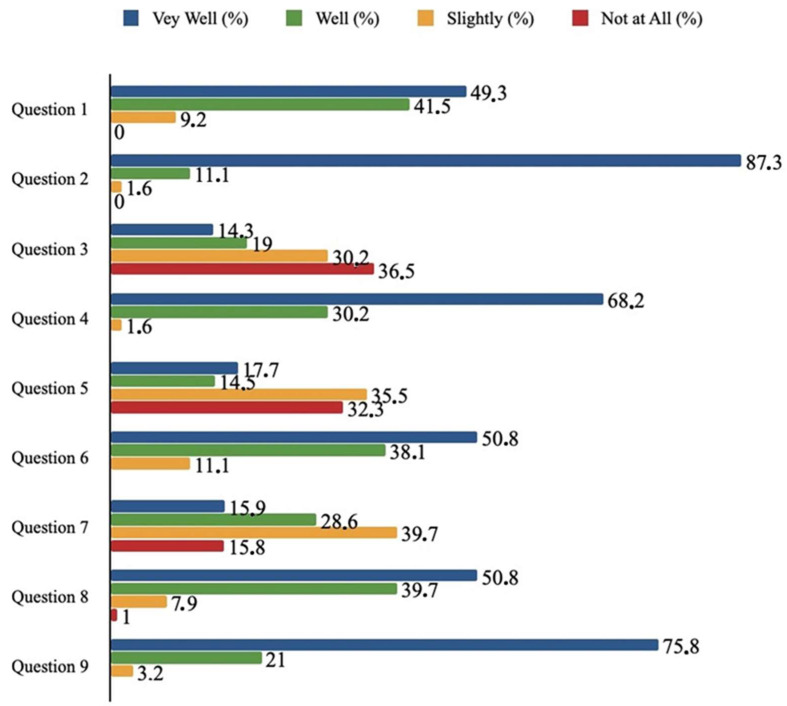
Responses to the questionnaire about the visualization of the laboratory results. Question 1: Each test is accompanied by a short explanatory sentence regarding its definition and role. Do you think it helped you to understand better its definition and role? Question 2: Each test is presented as a bullet bar with three colors: green (normal)—orange (borderline)—red (abnormal). Do you think it helped you to understand better how normal or how abnormal your test value is? Question 3: This format includes a test which is called non-HDL cholesterol. How well did you know its meaning? Question 4: Do you think the explanation of non-HDL cholesterol helped you understand its role and value? Question 5: This format includes a section with the title “Metabolic Syndrome Assessment”. How well did you know its existence? Question 6: Do you think the explanation and visualization of “Metabolic Syndrome Assessment” help you understand its role and value? Question 7: This format includes a section with the title “Insulin Resistant Assessment”. How well did you know its existence? Question 8: Do you think the explanation and visualization of “Insulin Resistance Assessment” help you understand its role and value? Question 9: Overall, do you think that our initiative has helped you gain a more complete picture of the usefulness of your tests and your metabolic profile?

**Table 1 life-14-01197-t001:** Information regarding the criteria was obtained from a questionnaire administered orally at the time of blood collection.

Exclusion Criteria	Inclusion Criteria
Individuals for whom the requested tests did not include the main parameters of an annual check-up	Individuals who presented for annual check-ups
Individuals with type 1 diabetes	Individuals with arterial hypertension
Individuals with autoimmune or chronic inflammatory diseases	Individuals with acquired dyslipidemia
Individuals with any form of malignancy within the last 3 years	Individuals with type 2 diabetes or prediabetes
Individuals with familial dyslipidemias or inherited metabolic disorders	Individuals with Hashimoto’s thyroiditis or non-immune hypothyroidism with normal TSH and fT4 levels
Pregnant and lactating women	Individuals with reported osteoarthritis or gout, either under treatment or in remission
Elderly women with advanced stage 4 osteoporosis (note: this information was available only from women who had recently undergone a BMD test)	Individuals receiving mild anxiolytic medication
Women with PCOS	Individuals with a history of fully treated malignancy for more than 3 years
Individuals with alcohol abuse problems	Women with a history of pregnancy lasting more than 6 months who are not breastfeeding
Individuals with viral or bacterial infections in the last 7 days	Individuals who have not had COVID-19 within the las t2 months
Individuals who had contracted COVID-19 in the last 3 months or individuals diagnosed with long COVID syndrome at the time of blood collection	Individuals aged 18–84 years of both genders
Elderly individuals aged > 85 years, due to their reduced social engagement and adaptability to daily life, as well as their increased morbidity and mortality	Individuals with reported social alcohol consumption (<2–3 drinks per week)
Individuals diagnosed with dementia, Alzheimer’s disease, or a serious mental illness	Individuals with a history of stroke, heart attack, pulmonary embolism, or deep vein thrombosis at least three months prior
Individuals with severe vascular complications in the last trimester	Individuals with none to moderate LTPA
Bedridden individuals	
Individuals under 17 years old	
Individuals engaged in strenuous physical activity due to their occupation	
Individuals engaging in moderate to vigorous leisure-time physical activities (LTPA), defined as either intense physical exercise three times a week or mild to moderate activities more than five times a week	
Individuals with very short stature (<160 cm in men and <150 cm in women) or very tall stature (>183 cm in men and >170 cm in women), as they may either underestimate or overestimate BMI	

**Table 3 life-14-01197-t003:** Demographics and medical characteristics of the study population after the use of inclusion and exclusion criteria.

	Selected Population	MetS (+)	BMI < 25 MetS (−)	BMI 25–30 MetS (−)	BMI > 30 MetS (−)
Number of Participants	1870	304	748	584	232
Men/Women (%)	42.6/57.4	55.3/44.7	40.1/59.9	44.9/55.1	37.8/62.2
Smoking (%)	32.1	30.9	31.5	35.0	28.3
Lipid-lowering therapy (%)	36.3	75	22.6	37.1	28.2
Hypothyroidism (%)	21.1	25	18.4	19.5	29.6
Antihypertensive therapy (%)	25.8	68.4	13.2	22.7	19.3
SD type 2	27.1	80.2	12.3	23.4	16.3

**Table 4 life-14-01197-t004:** Biochemical and Hematological parameters for each BMI category and for MetS group.

	MetS (+)	BMI < 25 MetS (−)	BMI 25–30 MetS (−)	BMI > 30 MetS (−)
Μedian + interquartile range (IQR)	Parameters as Markers of Cardiovascular Risk			
Total cholesterol (mg/dL)	177 (151–208)	184 (162–211)	190 (162–214)	187 (165–211)
Triglycerides (mg/dL)	148 (117–187)	72 (55–99)	92 (70–120)	95 (74–116)
HDL cholesterol (mg/dL)	44.5 (37.6–49.8)	61.1 (52.2–70.4)	55 (46.6–63.0)	53.4 (46.5–61.4)
LDL cholesterol (mg/dL)	96.6 (74.6–128.7)	104.8 (86.4–128.1)	111.6 (88.5–138.2)	113.4 (92.9–135.0)
Non-HDL cholesterol (mg/dL)	128.6 (104.1–162.6)	121.5 (101.4–146.2)	131.1 (105.6–158.7)	131.2 (111.4–156.2)
Cholesterol/HDL ratio	3.98 (3.23–5.01)	2.98 (2.57–3.50)	3.36 (2.81–4.16)	3.44 (2.94–3.96)
Non-HDL/HDL ratio	2.98 (2.23–4.01)	1.98 (1.58–2.50)	2.36 (1.81–3.16)	2.44 (1.94–2.96)
ALT/AST ratio	1.13 (0.89–1.41)	0.87 (0.71–1.04)	1.00 (0.79–1.21)	1.06 (0.86–1.26
	Parameters as Markers of Inflammation			
White Blood Cell count/μL	7.43 (6.10–8.77)	6.01 (5.16–7.05)	6.33 (5.48–7.45)	6.81 (5.73–7.63)
Neutrophils count/μL	4.14 (3.30–5.03)	3.20 (2.61–3.97)	3.45 (2.85–4.20)	3.75 (3.03–4.53)
Lymphocytes count/μL	2.35 (1.91–2.89)	2.11 (1.72–2.52)	2.18 (1.78–2.56)	2.23 (1.80–2.77)
Neutrophils/Lymphocytes Index (NLR Index)	1.78 (1.36–2.17)	1.52 (1.19–1.97)	1.60 (1.30–2.03)	1.72 (1.33–2.06)
	Parameters as common Markers for both Cardiovascular Risk and Inflammation			
Neutrophil/HDL ratio (NHR Index)	95.8 (71.3–125.0)	52.7 (39.7–70.2)	63.1 (48.0–85.2)	69.5 (54.9–91.1)
hs-CRP (mg/dL)	0.22 (0.10–0.53)	0.06 (0.03–0.15)	0.12 (0.05–0.24)	0.26 (0.11–0.61)
	Parameters as Markers of Glycemic Status			
Glucose	108 (101–118)	91 (86–97)	95 (88–101)	94 (89–99)
Triglycerides/HDL ratio	3.35 (2.51–4.68)	1.19 (0.86–1.74)	1.69 (1.14–2.36)	1.73 (1.31–2.32)
TyG Index	4.83 (4.73–4.99)	4.39 (4.35–4.57)	4.55 (4.38–4.69)	4.56 (4.45–4.66)
TyG-BMI Index	151.2 (136.9–171.1)	99.0 (90.7–105.8)	123.3 (115.8–129.8)	147.1 (138.4–161.6)
Insulin (mU/L)	14.29 (11.31–18.92)	7.64 (5.91–9.89)	9.78 (7.46–12.74)	12.38 (10.02–14.89)

**Table 5 life-14-01197-t005:** Biochemical and hematological parameters of MetS group and the statistical comparison with BMI groups (Wilcoxon signed-rank test).

	MetS (+) vs. BMI < 25 MetS (−)	MetS (+) vs. BMI 25–30 MetS (−)	MetS (+) vs. BMI > 30 MetS (−)
	Parameters as Markers of Cardiovascular Risk		
Total cholesterol	<0.001	0.009	0.023
Triglycerides	<0.001	<0.001	<0.001
HDL cholesterol	<0.001	<0.001	<0.001
LDL cholesterol	0.001	0.002	0.002
non-HDL cholesterol	0.351	0.397	0.752
Cholesterol/HDL ratio	<0.001	<0.001	<0.001
Non-HDL/HDL ratio	<0.001	<0.001	<0.001
ALT/AST ratio	<0.001	<0.001	<0.001
Neutrophil/HDL ratio (NHR Index)	<0.001	<0.001	<0.001
	Parameters as Markers of Inflammation		
White Blood Cell count	<0.001	<0.001	<0.001
Neutrophils count	<0.001	<0.001	<0.001
Lymphocytes count	<0.001	<0.001	0.06
Neutrophils/Lymphocytes Index (NLR Index)	<0.001	0.011	0.217
	Parameters as Common Markers for both Cardiovascular Risk and Inflammation		
Neutrophil/HDL ratio (NHR Index)	<0.001	<0.001	<0.001
hs-CRP	<0.001	<0.001	0.09
	Parameters as Markers of Glycemic Status		
Glucose	<0.001	<0.001	<0.001
Triglycerides/HDL ratio	<0.001	<0.001	<0.001
Insulin	<0.001	<0.001	<0.001

**Table 6 life-14-01197-t006:** Biochemical and hematological parameters of each BMI group and the statistical comparison between them (Wilcoxon signed-rank test).

	BMI < 25 MetS (−) vs. BMI 25–30 MetS (−)	BMI < 25 MetS (−) vs. BMI > 30 MetS (−)	BMI 25–30 MetS (−) vs. BMI > 30 MetS (−)
	Parameters as Markers of Cardiovascular Risk		
Total cholesterol	0.231	0.323	0.909
Triglycerides	<0.001	<0.001	0.123
HDL cholesterol	<0.001	<0.001	0.015
LDL cholesterol	0.641	0.162	0.577
Non-HDL cholesterol	0.054	0.103	0.465
Cholesterol/HDL ratio	<0.001	<0.001	0.117
Non-HDL/HDL ratio	<0.001	<0.001	0.116
ALT/AST ratio	<0.001	<0.001	<0.001
Neutrophil/HDL ratio (NHR Index)	<0.001	<0.001	<0.001
	Parameters as Markers of Inflammation		
White Blood Cell count	<0.001	<0.001	0.169
Neutrophils count	<0.001	<0.001	0.128
Lymphocytes count	0.015	0.002	0.213
Neutrophils/Lymphocytes Index (NLR Index)	0.367	0.135	0.589
	Parameters as Common Markers for both Cardiovascular Risk and Inflammation		
Neutrophil/HDL ratio (NHR Index)	<0.001	<0.001	<0.001
hs-CRP	<0.001	<0.001	<0.001
	Parameters as Markers of Glycemic Status		
Glucose	<0.001	0.084	0.815
Triglycerides/HDL ratio	<0.001	<0.001	0.045
Insulin	<0.001	<0.001	<0.001

**Table 7 life-14-01197-t007:** Comparing the performance of low-budget markers of IR in the participants without MetS with HOMA-IR as reference method (cut-off > 2.78).

	AUC	Cut-Off	Sensitivity (%)	Specificity (%)	Youden Index
TyG-BMI Index	0.77	122	65.6	77.1	0.43
TyG Index	0.72	4.57	59.4	73.1	0.32
Triglycerides/HDL ratio	0.69	1.88	51.5	75.3	0.27
ALT/AST ratio	0.65	1.06	50.4	71.9	0.22
NHR Index	0.65	6.98	50.5	70.5	0.22
NLR Index	0.57	1.83	40.1	70.3	0.11

**Table 8 life-14-01197-t008:** Comparison of TyG-BMI Index and the new PatrasLab Index in groups of participants without MetS with different IR prevalence. HOMA-IR (cut-off > 2.78) was used as reference method. IR was observed in 30% of all participants without MetS; thus, the categorization of training group 30% IR and evaluating group 30% was established. Evaluating groups with 20% and 10% IR were created after random exclusion of individuals with IR from the evaluating group 30%.

	Training Group 30% IR		Evaluating Group 10% IR		Evaluating Group 20% IR		Evaluating Group 30% IR	
	TyG-BMI Index	PatrasLab Index	TyG-BMI Index	PatrasLab Index	TyG-BMI Index	PatrasLab Index	TyG-BMI Index	PatrasLab Index
AUC	0.78	0.84	0.80	0.91	0.73	0.78	0.73	0.81
Cut-off	125	−0.49	130	−0.40	130	−0.50	121	−0.50
Sensitivity (%)	60.5	61.3	70.9	79.2	52.0	52.0	60.3	65.4
Specificity (%)	80.0	85.6	88.1	89.9	87.5	89.9	77.9	86.7
Youden Index	0.40	0.47	0.59	0.66	0.39	0.41	0.38	0.52
PPV (%)	56.3	65.1	39.6	46.8	50.0	56.5	53.9	68.2
NPV (%)	82.4	83.7	96.5	97.5	87.8	86.9	81.9	85.3
	Training group 30% IR		Evaluating group 10% IR		Evaluating group 20% IR		Evaluating group 30% IR	
	TyG-BMI Index + PatrasLab Index		TyG-BMI Index + PatrasLab Index		TyG-BMI Index + PatrasLab Index		TyG-BMI Index + PatrasLab Index	
* PPV (%)	84.1		86.1		83.3		84.8	

* Combined PPV.

## Data Availability

The original contributions presented in the study are included in the article/Appendix A. Further inquiries can be directed to the corresponding author.

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
