# Peer review of "Biomarkers and Data Visualization of Insulin Resistance and Metabolic Syndrome: An Applicable Approach"

_life, 2024, doi:10.3390/life14091197_

Round 1

Reviewer 1 Report

Comments and Suggestions for Authors

Thank you very much for the invitation to review the article " Biomarkers and Data Visualization of Insulin Resistance and Metabolic Syndrome: An Applicable Approach" Insulin Resistance and Metabolic Syndrome are two very important and relevant issues. Patients’ awareness of their results is critical for the management success. I would like to provide some comments on the various sections of the article.

ABSTRACT

Clarification is needed on how researchers determined that only two markers, "BMI and NHR index," are essential in assessing metabolic health patterns. This should be highlighted in the results, discussion, or conclusions sections.

INTRODUCTION

Including not only diabetes but all outcomes of insulin resistance (IR) and metabolic syndrome (MetS) would be beneficial. Describing the pros and cons of the diagnostic methods for IR and MetS used in clinical practice, possibly through a table, would be informative. Sharing the cut-off levels for different equations from different studies or criteria for IR and MetS and how these were established would also be helpful.

Transferring pertinent details from the Materials and Methods section to the Introduction could improve understanding.

Additionally, discussing other researchers' experiences and studies’ results on the importance of patients' awareness of their blood test results, in terms of normality or abnormality, is crucial, particularly concerning patient safety, self-care, and treatment adherence. Data on patient awareness and the outcomes of relevant studies, along with any gaps in data and emerging research questions, should be provided.

MATERIALS AND METHODS

Describing the study design, inclusion and exclusion criteria, patient informed consent prior to enrollment, providing a study flowchart, and detailing the statistical analysis used would be beneficial.

Each measurement utilized should be clearly defined. For instance, it is unclear how researchers handled two measurements – whether they selected the second one or the average of two for statistical analysis, as indicated by "measurements were taken twice." All laboratory measurements should include normal values, units, and the method of laboratory assessment. Each calculation or equation must be referenced. The PatrasLab Index equation should be included in the methods section. Examples of Data Visualization should be added as supplementary materials.

Results (basal data) should be moved to the Results section and presented in a structured format.

The groups "Training group 30% IR, Evaluating group 10% IR, Evaluating group 20% IR, Evaluating group 30% IR" need clarification in the methods section, as Table 5 is not clear. The statement from the Results section: "Subsequently, the indexes were tested in three variants of the control group with varying prevalence of insulin resistance (IR): low (10%), moderate (20%), and high (30%) IR prevalence," is questionable.

The responses to Data Visualization questions on the Likert scale require explanation, and the groups from Figure 1 should be described. It should be clarified whether these are the same participants who received questionnaires as in the main study.

RESULTS

Explaining the baseline data of all participants in the table, as well as the distinctions between the MetS (+) and BMI (-) groups, would be beneficial. It would also be helpful to illustrate the division into all groups that researchers analyzed and to indicate the number of participants with insulin resistance (IR) in both MetS (+) and BMI (-) groups according to the HOMA IR calculation (2.78), along with the variances of other IR equations within the IR groups.

Clarifying the meaning of the "HOMA IR" column in Table 4 is crucial. Describing Table 5 and the differences between groups is essential. Elucidating Figure 1 is necessary, as its significance and implications are not immediately apparent.

DISCUSSION

The discussion lacks references to literature and data from other studies to substantiate the results' discussion. Expanding the discussion to include findings on IR and patient awareness of blood test results would be advantageous.

The study's strengths and limitations should be articulated.

CONCLUSIONS

It is critical to elaborate and organize the conclusions in line with the results' explanation. Conclusions should detail the specific findings of the study and be consistent with the abstract.

Comments on the Quality of English Language

Minor editing of English language required.

Author Response

Comment 1: ABSTRACT. Clarification is needed on how researchers determined that only two markers, "BMI and NHR index," are essential in assessing metabolic health patterns. This should be highlighted in the results, discussion, or conclusions sections.

Thank you for your comment. The new information are added in lines 19-25 of the revised manuscript.

Comment 2: INTRODUCTION. Including not only diabetes but all outcomes of insulin resistance (IR) and metabolic syndrome (MetS) would be beneficial.

Thank you for your comment. We added the requested data for IR in lines 35-37 (Reference 1). The outcomes for MetS are already in lines 45-47 (References 2,6,7)

Comment 3: Describing the pros and cons of the diagnostic methods for IR and MetS used in clinical practice, possibly through a table, would be informative. Sharing the cut-off levels for different equations from different studies or criteria for IR and MetS and how these were established would also be helpful.

We agree with this comment. Regarding the pros and cons of the diagnostic methods for IR, we added lines 52-57. For MetS the two algorithms are already mentioned in Paragraph 2.6. The other data about the cut-offs are added in Table 2 of the revised manuscript.

Comment 4: Transferring pertinent details from the Materials and Methods section to the Introduction could improve understanding.

Thank you for your comment. We have transferred some details according to the other reviewer’s comments, but we believe that the explanation of BMI is essential for the understanding of the study design.

Comment 5: Additionally, discussing other researchers' experiences and studies’ results on the importance of patients' awareness of their blood test results, in terms of normality or abnormality, is crucial, particularly concerning patient safety, self-care, and treatment adherence. Data on patient awareness and the outcomes of relevant studies, along with any gaps in data and emerging research questions, should be provided.

Thank you for your comment. Lines 65-66 were added

Comment 6: MATERIALS AND METHODS. Describing the study design, inclusion and exclusion criteria, patient informed consent prior to enrollment, providing a study flowchart, and detailing the statistical analysis used would be beneficial.

Thank you for your comment. Lines 74-87 (paragraph 2.1), 96-103(paragraph 2.2), Table 1 and Figure 1 were added. Also, lines 491-492 about patient informed consent and lines 245-249 detailing the statistical analysis were added

Comment 7: Each measurement utilized should be clearly defined. For instance, it is unclear how researchers handled two measurements – whether they selected the second one or the average of two for statistical analysis, as indicated by "measurements were taken twice." All laboratory measurements should include normal values, units, and the method of laboratory assessment.

Thank you for your comment. The sentence "measurements were taken twice." Was changed as  “Measurements were taken twice and the mean value was chosen.” (lines 136-137). Table 2 was added.

Comment 8: Each calculation or equation must be referenced. The PatrasLab Index equation should be included in the methods section.

Thank you for your comment. Lines 196-197, 205-217 and paragraph 2.9 were added.

Comment 9: Examples of Data Visualization should be added as supplementary materials.

Thank you for your comment. We have added figures of Data Visualization as supplementary materials.

Comment 10: Results (basal data) should be moved to the Results section and presented in a structured format.

Thank you for your comment. Paragraph 3.1 was added.

Comment 11: The groups "Training group 30% IR, Evaluating group 10% IR, Evaluating group 20% IR, Evaluating group 30% IR" need clarification in the methods section, as Table 5 is not clear. The statement from the Results section: "Subsequently, the indexes were tested in three variants of the control group with varying prevalence of insulin resistance (IR): low (10%), moderate (20%), and high (30%) IR prevalence," is questionable.

Thank you for your comment. The clarification is in chapter 2.9 and Table 8 of the revised manuscript.

Comment 12: The responses to Data Visualization questions on the Likert scale require explanation, and the groups from Figure 1 should be described. It should be clarified whether these are the same participants who received questionnaires as in the main study.

Thank you for this comment. However, we believe that the responses to Data Visualization are reported sufficiently, please clarify any extra data that should be reported.

Comment 13: RESULTS. Explaining the baseline data of all participants in the table, as well as the distinctions between the MetS (+) and BMI (-) groups, would be beneficial. It would also be helpful to illustrate the division into all groups that researchers analyzed and to indicate the number of participants with insulin resistance (IR) in both MetS (+) and BMI (-) groups according to the HOMA IR calculation (2.78), along with the variances of other IR equations within the IR groups.

Thank you for your comment. Paragraph 3.1, lines 312-314 and Table 3 were added. Also, changes on Table 1 (Table 4 in the revised manuscript) were added.

Comment 14: Clarifying the meaning of the "HOMA IR" column in Table 4 is crucial.

We agree with this comment, so we changed the Table to Table 7 of the revised manuscript.

Comment 15: Describing Table 5 and the differences between groups is essential.

Description of the Table (now Table 8 in the revised manuscript) was added in line 343-348.

Comment 16: Elucidating Figure 1 is necessary, as its significance and implications are not immediately apparent.

Thank you for this comment, however we believe that the explanation of the Figure is sufficient.

Comment 17: DISCUSSION. The discussion lacks references to literature and data from other studies to substantiate the results' discussion. Expanding the discussion to include findings on IR and patient awareness of blood test results would be advantageous.

Thank you for this comment. Lines 391-402 and 434-437 were added.

Comment 18: The study's strengths and limitations should be articulated.

Thank you for this comment. The study’s strengths are reported in lines 398-400, 411-413, 427-428, 431-433. Also, limitations of the study were added  (Lines 466-482)

Comment 19: CONCLUSIONS. It is critical to elaborate and organize the conclusions in line with the results' explanation. Conclusions should detail the specific findings of the study and be consistent with the abstract.

Thank you for this comment. We changed the conclusions section so that the findings about MetS, IR, NHR Index and PatrasLab Index are reported first data visualization later (Lines 245-249).

Reviewer 2 Report

Comments and Suggestions for Authors

This cross-sectional study aims to evaluate low-budget biochemical and hematological parameters, with data visualization, for identifying insulin resistance and metabolic syndrome in a community-based laboratory, considering that annual check-ups are inadequate for early detection. They included 1870 participants in Patras, Greece, and blood samples were analyzed for key cardiovascular and inflammatory markers. Diagnostic markers for insulin resistance (TyG-Index, TyG-BMI, Triglycerides/HDL ratio, NLR) were compared with HOMA-IR. They concluded that BMI and NHR index could be essential in assessing metabolic health patterns. There are numerous issues in the methodology section:

The issues to be resolved:

1. Please, line 35, it should be type 2 diabetes instead of “Diabetes Mellitus”

2. Line 92. Please add more information about the Institution where the recruitment of the subjects was done, and what level of medical care is that

3. Inclusion and exclusion criteria, please add

4. Research design should be more clear

5. Line 124, please add kg/m2

6. Please add a more precisely cut off for the HOMA IR value

7. Line 173, please add TyG index explanation, and put the reference

8. Lines 193-200, please add Table and numeric data

9. Please add units for all parameters in Table 1

10,  Please add limitations of the study

11. Lines 340-343 with references, should be in the discussion section  

Author Response

Comment 1. Please, line 35, it should be type 2 diabetes instead of “Diabetes Mellitus”

Thank you for the comment. We added the change (line 38 in the revised manuscript).

Comment 2. Line 92. Please add more information about the Institution where the recruitment of the subjects was done, and what level of medical care is that

Thank you for the comment. We added this information in lines 141-142.

Comment 3. Inclusion and exclusion criteria, please add

We added this information in Table 1.

Comment 4. Research design should be more clear

We agree with this comment. Study design was analyzed (lines 74-87) and a flowchart was added as Figure 1.

Comment 5. Line 124, please add kg/m2

The information was added (lines 266-267 in the revised manuscript).

Comment 6. Please add a more precisely cut off for the HOMA IR value

In Paragraph 3.3 of the Results section we state that the 90th percentile of healthy individuals with BMI< 25 is used as cut-off, according to the reference 38.

Comment 7. Line 173, please add TyG index explanation, and put the reference

Thank you for the comment. We added the data and reference 34 in Table 2.

Comment 8. Lines 193-200, please add Table and numeric data

Thank you for the comment. However, there are too many data to mention in the text, so we summarize them at the beginning of Paragraph 3.1 of the Results section (lines 253-271). 

Comment 9. Please add units for all parameters in Table 1

We have already added units where they apply in the Table

Comment 10.  Please add limitations of the study

We agree with this comment. We have added a paragraph with the requested limitations after the Conclusions (lines 466-482).

Comment 11. Lines 340-343 with references, should be in the discussion section

 These lines were moved at the end of the discussion section.

Reviewer 3 Report

Comments and Suggestions for Authors

This study is of reasonably high quality, and I think the introduction, presentation of results, and discussion are generally well done. However, due to a lack of novelty, population bias, and insufficient test data, I believe it does not meet the standards required for publication in Life.

-The exclusion criteria are ambiguous. It is essential to specify exactly which individuals were excluded and how many of them were excluded.

-It is mentioned that some patients refused to have their waist circumference measured, but what percentage of the total does this represent? In many diagnostic criteria for metabolic syndrome, waist circumference is considered a crucial criterion, as it highly sensitively reflects visceral fat levels, which in turn indicate the presence of insulin resistance. I believe it would be better to conduct the analysis excluding those in this study's sample who refused to have their waist circumference measured.

-It is very interesting that people with insulin resistance have high TyG-BMI values, but this has already been thoroughly proven in numerous studies, and it lacks novelty.

-It is stated that 21.1% have Hashimoto's thyroiditis or non-immune hypothyroidism. Isn't this rate too high compared to the general population?

Comments on the Quality of English Language

This study is of reasonably high quality, and I think the introduction, presentation of results, and discussion are generally well done. However, due to a lack of novelty, population bias, and insufficient test data, I believe it does not meet the standards required for publication in Life.

Author Response

Comment 1. This study is of reasonably high quality, and I think the introduction, presentation of results, and discussion are generally well done. However, due to a lack of novelty, population bias, and insufficient test data, I believe it does not meet the standards required for publication in Life.

Thank you for your comment. However, we believe that our study does not lack novelty.  To our knowledge, this is the first study of the performance of IR indexes in such a big sample of the Greek population. Furthermore, it is the first time that NHR index, an inflammation index, is included in equations and models for the diagnosis of MetS and IR, two pathologic situations with a background of inflammation, Lastly, this is the first study of the response to data visualization of blood results in the community.

Comment 2. The exclusion criteria are ambiguous. It is essential to specify exactly which individuals were excluded and how many of them were excluded.

Thank you for your comment. The information was added in Table 1 and Figure 1.

Comment 3. It is mentioned that some patients refused to have their waist circumference measured, but what percentage of the total does this represent? In many diagnostic criteria for metabolic syndrome, waist circumference is considered a crucial criterion, as it highly sensitively reflects visceral fat levels, which in turn indicate the presence of insulin resistance. I believe it would be better to conduct the analysis excluding those in this study's sample who refused to have their waist circumference measured.

Thank you for your comment. As already stated, this was the case at the beginning of our study, as later waist circumference measurement was abandoned, so the number of those individuals is low (approximately 200 individuals). We have included this I the limitations of the study (line 466 of the the revised manuscript).

Comment 4. It is very interesting that people with insulin resistance have high TyG-BMI values, but this has already been thoroughly proven in numerous studies, and it lacks novelty.

Thank you for your comment. However, this is the first study of the Greek population and there are only a few studies of the Caucasian race. Furthermore, we consider the fact that the cut-off was found to be quite different from the international bibliography, an interesting finding.

Comment 5. It is stated that 21.1% have Hashimoto's thyroiditis or non-immune hypothyroidism. Isn't this rate too high compared to the general population?

We agree with this comment, considering that the prevalence of Hashimoto's thyroiditis or non-immune hypothyroidism is 9% in the Greek population and 14.4% in women. We believe that this is the case because it was in the inclusion criteria and because individuals with hypothyroidism are more likely to come for an annual check-up than healthy people.

Reviewer 4 Report

Comments and Suggestions for Authors

This paper used data visualization and descriptive analysis to identify insulin resistance and metabolic syndrome among approximately 187 participants in Greece. The findings suggest that data visualization, along with two specific attributes—BMI and NHR—can effectively assess health patterns. The paper is well-written, but several improvements could be made:

- Follow the journal instructions carefully: https://www.mdpi.com/journal/life/instructions. 

- Remove headings like background, methods, etc., from the abstract.

- Spell out abbreviations such as BMI and NHR at their first appearance in both the abstract and the manuscript.

- Present the summary of contributions at the end of the introduction section.

- Consider adding a literature review or related studies section between the introduction and materials section to improve flow and highlight previous research on diabetes/prediabetes detection.

- Including a table that summarizes and compares previous works with this study would enhance its contribution.

- A flowchart in section 2 would help guide readers through the study’s methodology.

- The Helsinki statement related to the ethics board, including the approval number, should be included in section 2, as the study involves human participants.

- Present the profiles of 486 individuals, grouped by age range, education, job, etc., to provide context on the respondents' backgrounds.

- Consider incorporating these related papers on diabetes research to enhance the descriptive methods: https://doi.org/10.3390/math10214027, https://doi.org/10.3390/math11102266.

- In the conclusion, provide a brief summary of the results and their implications. The current version only mentions the need for future research. Also, consider elaborating on some limitations of the study.

Comments on the Quality of English Language

Final english check is required.

Author Response

Comment 1. Follow the journal instructions carefully: https://www.mdpi.com/journal/life/instructions. Remove headings like background, methods, etc., from the abstract.

Thank you for this comment. We made the necessary changes in the abstract.

Comment 2. Spell out abbreviations such as BMI and NHR at their first appearance in both the abstract and the manuscript.

Thank you for this comment. We added the information in lines 77 and 181.

Comment 3. Present the summary of contributions at the end of the introduction section.

Author contributions are stated in lines 483-485 according to the guidelines.

Comment 4. Consider adding a literature review or related studies section between the introduction and materials section to improve flow and highlight previous research on diabetes/prediabetes detection.

Thank you for this comment. We added references 1, 10, 11, 12.

Comment 5. Including a table that summarizes and compares previous works with this study would enhance its contribution.

Table 2 was added.

Comment 6. A flowchart in section 2 would help guide readers through the study’s methodology.

We agree with this comment. A flowchart was added as Figure 1.

Comment 7. The Helsinki statement related to the ethics board, including the approval number, should be included in section 2, as the study involves human participants.

Thank you for this comment. We added this information in paragraph 2.4

Comment 8.  Present the profiles of 486 individuals, grouped by age range, education, job, etc., to provide context on the respondents' backgrounds.

Thank you for this comment. The information we have is our decision to send the questionnaire to individuals 18-50 years old who use e-mail on daily basis. The answers were anonymous (via google form).

Comment 9. Consider incorporating these related papers on diabetes research to enhance the descriptive methods: https://doi.org/10.3390/math10214027, https://doi.org/10.3390/math11102266.

References 1 and 52 were added.

Comment 10. In the conclusion, provide a brief summary of the results and their implications. The current version only mentions the need for future research. Also, consider elaborating on some limitations of the study.

We have added extra information in lines 466-482.

Round 2

Reviewer 2 Report

Comments and Suggestions for Authors

The authors corrected the manuscript according to the suggestions, I have no further unresolved issues.

Reviewer 3 Report

Comments and Suggestions for Authors

I believe the study lacks novelty, but the manuscript itself is correctly structured. I have no further revision requests. The decision on whether to publish it in Life is up to the editorial office.